# Sharing Is Caring—Data Sharing Initiatives in Healthcare

**DOI:** 10.3390/ijerph17093046

**Published:** 2020-04-27

**Authors:** Tim Hulsen

**Affiliations:** Department of Professional Health Solutions & Services, Philips Research, 5656AE Eindhoven, The Netherlands; tim.hulsen@philips.com

**Keywords:** data sharing, data management, data science, big data, healthcare

## Abstract

In recent years, more and more health data are being generated. These data come not only from professional health systems, but also from wearable devices. All these ‘big data’ put together can be utilized to optimize treatments for each unique patient (‘precision medicine’). For this to be possible, it is necessary that hospitals, academia and industry work together to bridge the ‘valley of death’ of translational medicine. However, hospitals and academia often are reluctant to share their data with other parties, even though the patient is actually the owner of his/her own health data. Academic hospitals usually invest a lot of time in setting up clinical trials and collecting data, and want to be the first ones to publish papers on this data. There are some publicly available datasets, but these are usually only shared after study (and publication) completion, which means a severe delay of months or even years before others can analyse the data. One solution is to incentivize the hospitals to share their data with (other) academic institutes and the industry. Here, we show an analysis of the current literature around data sharing, and we discuss five aspects of data sharing in the medical domain: publisher requirements, data ownership, growing support for data sharing, data sharing initiatives and how the use of federated data might be a solution. We also discuss some potential future developments around data sharing, such as medical crowdsourcing and data generalists.

## 1. Introduction

The past years have seen a steep rise in the amount of health data being generated. These data come not only from professional health systems (MRI scanners, pathology slides, DNA tests, etc.) but also from wearable devices. All these data combined form ‘big data’ that can be utilized to optimize treatments for each unique patient (‘precision medicine’) [1]. To achieve this precision medicine, it is necessary that hospitals, academia and industry work together to bridge the ‘valley of death’ of translational medicine [2]. However, hospitals and academia often have problems with sharing their data, even though the patient is actually the owner of his/her own health data, and data sharing is associated with increased citation rate [3,4]. Academic hospitals usually want to be the first ones to publish papers on the data, because they spent a lot of time in setting up clinical trials and collecting the data. Society benefits the most if the patient’s data are shared as soon as possible so that other researchers can work with it [5], but this idea has not settled in yet. Some datasets are publicly available (e.g., in prostate cancer [6]), but these are usually only shared after studies are finished and/or publications have been written based on the data, which means a severe delay of months or even years before others can use the data for analysis. One solution is to incentivize the hospitals [7,8] to share their data with (other) academic institutes and the industry. Besides this academic reluctance, data is also being shared less because of stricter privacy laws such as the EU General Data Protection Regulation (GDPR) [9] and the California Consumer Privacy Act (CCPA) [10]. At the moment, only around 10% of the world’s population has it personal information covered by the GDPR or similar laws, but Gartner Research predicts that this will be around 50% by 2022 [11]. There is an increasingly urgent need to balance the opportunity big data provides for improving healthcare, against the right of individuals to control their own data [1]. Scientists should maximize their efforts to improve healthcare, but they should also only use data with appropriate informed consent. This open science vs. privacy balance will remain an increasing challenge for the coming years.

The topic of data sharing has received more attention in recent years. In 1980, only 46 articles (0.0186% of the total) published in PubMed contained the keyword “data sharing”, while in 2019 there were 5960 articles (0.4253% of the total) containing this keyword (Figure 1). It is also interesting to see the sudden rise of interest in the subject since 2016, the year of the approval of the GDPR, and another peak in 2018, the year of its enforcement.

If we use PubMed to find terms related to “data sharing”, there are some interesting observations (Figure 2). Mostly used are obviously terms such as “patients”, “health”, “study” and “information”, but closely behind these are “use” (or “used”/”using”), “treatment”, “care” “analysis” and “rights”. “Use” might point to the fact that data collection and sharing is closely connected to the usage of the data, i.e., in the consent form it should be mentioned in detail what the health data will be used for. “Treatment”, “care” and “analysis” point to one of the main uses of the data: analysis in order to improve treatment and care, for example in clinical decision support (CDS) systems. “Rights” is probably related to the patients’ privacy rights when it comes to data sharing, an issue that is discussed in detail in this manuscript.

There have been some studies on the conditions and challenges for sharing data. For example, for the BigData@Heart platform of the Innovative Medicines Initiative (IMI), a descriptive case study into the condition for data sharing was carried out [12]. Principle investigators of the participating databases were requested to send any kind of documentation that possibly specified the conditions for data sharing, which were then qualitatively reviewed for conditions related to data sharing and data access. This review revealed overlap on the conditions: (1) only to share health data for scientific research, (2) in anonymized/coded form, (3) after approval from a designated review committee, and while (4) observing all appropriate measures for data security and in compliance with the applicable laws and regulations. These challenges give thought to the design of an ethical governance framework for data sharing platforms. The conclusion of the case study was that current data sharing initiatives should concentrate on: (1) the scope of the research questions that may be addressed, (2) how to deal with varying levels of de-identification, (3) determining when and how review committees should come into play, (4) align what policies and regulations mean by “data sharing” and (5) how to deal with datasets that have no system in place for data sharing.

Sharing data should not just be a one-way street from the clinician to the researcher; ideally the clinician, the researcher and the patient (or patient organization) would work together on setting up the study, so that there is an agreement on data usage upfront, and expectations are managed. Sharing data will also increase confidence and trust in the conclusions drawn from clinical trials [14]. It will help to enable the independent confirmation of results (reproducibility), an essential part of the scientific process. It will foster the development and testing of new hypotheses. Sharing clinical trial data should also make progress more efficient by making the most of what may be learned from each trial and by avoiding unwarranted repetition. It will help to satisfy the moral obligation of researchers towards study participants, and it will benefit patients, investigators, sponsors, and society. In this review, we discuss several aspects of data sharing in the medical domain. The Section 2 is about publisher requirements, which shows what guidelines have been created by publishers and editors to promote the sharing of data. Since academics rely on publication of their data, these are important measures and a logical first topic to be discussed. The Section 3 shows that there is an ongoing discussion about data ownership, which influences the way that regulations are being implemented. The Section 4 shows the growing support for data sharing, making the link to open science and the reproducibility of results. The Section 5 shows data sharing initiatives that have been undertaken recently. The Section 6 and Section 7 discusses how the use of federated data might be a solution of the privacy and reproducibility issues mentioned in the Section 2, Section 3 and Section 4.

## 2. Publisher Requirements

Most publishers strongly recommend sharing research data. For this section, the publisher requirements of five major publishers are discussed, as well as the most widely used sets of guidelines from publishers and editors.

Nature states that data sharing makes new types of research possible [15], for example through the pooling of patient cohorts, and hints to future developments: sharing data is not only a way to improve the reproducibility and robustness of the science that is taking place today, but can drive new science for tomorrow. By browsing through existing datasets, new hypotheses can be formed, which can then be tested in new studies. Because nobody can predict how valuable a dataset will be in the future, data should be made available to future scientists whenever possible. The Science journals support the efforts of databases that aggregate published data for the use of the scientific community [16]. Therefore, before publication, large data sets must be deposited in an approved database and an accession number or a specific access address must be included in the published paper. The Science journals also encourage compliance with Minimum Information for Biological and Biomedical Investigations (MIBBI) guidelines [17]. British Medical Journal (BMJ) journals have three different data sharing policies (“tiers”), dependent of the journal [18]. They encourage researchers to make available as much of the underlying data from an article as possible (without compromising the privacy of the patients). The BMJ journals also consider reproducibility: all data that are needed to reproduce the results presented in the associated article should be made available. When submitting a manuscript to a publisher such as BioMed Central (BMC), the researcher even “agrees to make the raw data and materials described in your manuscript freely available to any scientist wishing to use them for non-commercial purposes, as long as this does not breach participant confidentiality” [19]. Public Library of Science (PLOS) journals require authors “to make all data necessary to replicate their study’s findings publicly available without restriction at the time of publication. When specific legal or ethical restrictions prohibit public sharing of a data set, authors must indicate how others may obtain access to the data” [20]. Other publishers have similar guidelines in place, promoting data sharing on a global level.

In 2015, the Transparency and Openness Promotion (TOP) guidelines [21] were published. The guidelines were developed to translate scientific norms and values into concrete actions and change the current incentive structures to drive researchers’ behavior toward more openness. The TOP guidelines have eight standards: (1) citation standards; (2) data transparency; (3) analytics methods (code) transparency; (4) research materials transparency; (5) design and analysis transparency; (6) preregistration of studies; (7) preregistration of analysis plans; and (8) replication. For each standard, there are three levels with increasing stringency. Currently, over 1000 scientific journals have implemented the TOP guidelines [22].

The International Committee of Medical Journal Editors (ICMJE) also recommends the sharing of data [14]. In 2016, they proposed to require authors to share with others the deidentified individual-patient data (IPD) underlying the results presented in the article no later than 6 months after publication. The data underlying the results are defined as “the IPD required to reproduce the article’s findings, including necessary metadata”. Since 2019, the ICMJE requires investigators to register a data-sharing plan when registering a trial as well. This plan must include where the researchers will house the data and, if not in a public repository, the mechanism by which they will provide others access to the data, whether data will be freely available to anyone upon request or only after application to and approval by a learned intermediary, whether a data use agreement will be required, etc. Declaring the plan for sharing data prior to their collection will further enhance transparency in the conduct and reporting of clinical trials by exposing when data availability following trial completion differs from prior commitments. However, ICMJE also stresses that the rights of investigators and trial sponsors must be protected. To achieve this, the following four rules apply: (1) editors will not consider the deposition of data in a registry to constitute prior publication; (2) authors of secondary analyses using these shared data must attest that their use was in accordance with the terms (if any) agreed to upon their receipt; (3) authors of secondary analyses must reference the source of the data using a unique identifier of a clinical trial’s data set to provide appropriate credit to those who generated it and allow searching for the studies it has supported; (4) authors of secondary analyses must explain completely how theirs differ from previous analyses. In addition, those who generate and then share clinical trial data sets deserve substantial credit for their efforts. Those using data collected by others should seek collaboration with those who collected the data.

By providing the guidelines and rules set out above, the publishers and editors contribute to the acceptance of data sharing by researchers. Not only does it help solve their problem of a lack of reproducibility of the scientific results published in their journals, increasing confidence and trust in these results; it will also help the scientists in the generation of new hypotheses, and avoiding unnecessary repetition. In the end, publishers, as well as scientists, patients and societies will benefit from complying with these rules.

## 3. Data Ownership

When discussing the sharing of data, it is important to realize that there is not much consensus on who is actually the owner of that data. This section briefly discusses this issue of data ownership in the light of recent privacy laws. These laws have a very large impact on the topic of data sharing.

Institutions tend to believe that they own the patient data, since they collected it. However, these institutions are in fact just “data custodians”; the data is the property of the patient and the access and use of that data outside of the clinical institute usually requires patient consent [1]. This limits the exploitation of the “big data” that are available in the clinical records, because the data should be destroyed (or sufficiently anonymized) after the end of the study. Big data techniques such as machine learning and deep learning use thousands to millions of data points, which may have required considerable processing. It would be a waste to lose such valuable data at the end of the project. Therefore, it is advised to ask the patient for consent to store and use their data for future scientific research. Although it is not possible to use the data from a large number of retrospective datasets in this manner, this will make sure that at least the prospectively collected data can be used in future studies. The dilemma of the use of patient data versus privacy rights has gotten much attention because of the implementation of the GDPR in 2018 (as well as the CCPA in 2020), initiating an international debate on the sharing of big data in the healthcare domain [23]. Earlier laws such as the Health Insurance Portability and Accountability Act (HIPAA) Privacy Rule [24] of the USA and the Personal Information Protection and Electronic Documents Act (PIPEDA) [25] of Canada already gave more rights to patients regarding their data, but the GDPR and CCPA have taken it to another level. However, GDPR and similar laws do not say much about data ownership. The GDPR’s main entities are the data controller and the data processor [9]. “Data controller” means the natural or legal person, public authority, agency or other body which, alone or jointly with others, determines the purposes and means of the processing of personal data. “Data processor” means a natural or legal person, public authority, agency or other body which processes personal data on behalf of the controller. In countries outside of the European Union, where GDPR does not apply, there is also not much agreement on data ownership, making it even more justifiable to always ask for the consent of the patient.

## 4. Growing Support for Data Sharing

The idea that data should be shared as much as possible to enable scientific progress is gaining momentum, mostly because of the power of big data analyses, machine learning, deep learning, etc. In this section, some developments are discussed which show this growing support for data sharing. Some of them were already known to the author, whereas others were a result from the literature analysis mentioned in the introduction.

Science in Transition [26] claims that “science has become a self-referential system where quality is measured mostly in bibliometric parameters and where societal relevance is undervalued”, emphasizing that researchers tend to care mostly about publications instead of using the data to solve real-life problems. It also gives attention to the reproducibility problem in science: more than 70% of researchers have tried and failed to reproduce another scientist’s experiments, and more than half have even failed to reproduce their own experiments [27]. This problem is not only caused by a lack of data sharing, but also because researchers do not share methodologies used to combine and analyse datasets. In many projects, data from several sources (possibly collected using different protocols and standards) need to be combined before the data analysis can take place. If these methodologies, as well as the analysis scripts, are not shared, results cannot be reproduced even if the data is available. This reproducibility issue could be resolved by ‘Open Science’, which is defined as the practising of science in a sustainable manner which gives others the opportunity to work with, contribute to and make use of the scientific process. This allows users from outside science to influence the research world with questions and ideas and help gather research data [28]. The Open Science movement stimulates not only open access to data, but also open access publishing, open source scientific software and open educational resources [29].

The Mayo Clinic Platform [30] is a new cloud-based clinical data analytics platform, storing de-identified patient data, which providers, payers and pharmaceutical companies outside of Mayo can link up to via application programming interfaces (APIs), as well as establishing standard templates for compliance and legal agreements. The first partner of the Mayo Clinic Platform is Nference, a software startup that Mayo is an investor in. Nference develops analytics, machine learning and natural language processing tools that “augment” the work of data scientists, in order to help research organizations and pharmaceutical companies conduct “research at scale”. Mayo Clinic hopes to work with pharma to commercialize new therapies. Mayo itself wouldn’t commercialize those therapies, though the system could receive royalties from insights generated on the platform. These royalties would be re-invested into Mayo’s clinical practice, research and education work.

Healthcare Business and Technology wrote about how data sharing could change the entire healthcare industry [31]. It discusses the partnership announced by Apple in 2018 with 13 major healthcare systems, including Johns Hopkins and the University of Pennsylvania, that will allow Apple to download patients’ electronic health data onto its devices (with consent of the patients). This type of data sharing could transform the U.S. healthcare industry by empowering patients in new ways and improving care. It could even reduce organizational costs by streamlining care processes, because hospital staff would need to spend less time on making data available to patients. And artificial intelligence (AI) could use the patient data to answer patients’ questions and direct them to the healthcare services they need.

The ‘Ten Commandments of Translational Research Informatics’ [32] are some guidelines related to data management and data integration in translational research projects. Some of the commandments relate to the sharing of data: clear arrangements about data access need to be made (commandment 4), agree about de-identification and anonymization (commandment 5), the FAIR guiding principles [33] should be adhered to (commandment 8), and researchers should think about what will happen to the data after the project (commandment 10): e.g., research can be shared in a public repository.

## 5. Data Sharing Initiatives

There are many initiatives around the world supporting the sharing of medical data, leading the way to open science while still respecting the privacy rights of the patients. This paragraph gives some recent examples of these initiatives, resulting from the literature analysis from the introduction.

GIFT-Cloud [34] is a platform for data sharing and collaboration in medical imaging research. The goal of GIFT-Cloud is to provide a flexible, clinician- and researcher-friendly system for anonymising and sharing data across multiple institutions. It was built to support the Guided Instrumentation for Fetal Therapy and Surgery (GIFT-Surg) project, an international research collaboration that is developing novel imaging methods for fetal surgery, but it also has general applicability to other areas of imaging research. It simplifies the transfer of imaging data from clinical to research institutions, facilitating the development and validation of medical research software and the sharing of results back to the clinical partners. GIFT-Cloud supports collaboration between multiple healthcare and research institutions while satisfying the demands of patient confidentiality, data security and data ownership. It achieves this by building upon existing, well-established cross-platform technologies. GIFT-Cloud stores data from each institution in a separate data group and access to these groups can be individually configured for each account, corresponding with what is arranged in the data sharing agreements. 

Another development in data sharing is the Personalized Consent Flow [35]: a new consent model that allows people to control their personally collected health data and determine to what extent they want to share these for research purposes. Three main features characterize the consent flow: (1) Users are asked general questions about sharing data. When they wish to share data for scientific research, they may opt for “narrow” consent (treating each study separately) or “broad” consent (for multiple studies). Furthermore, users can decide which data will be shared for specific studies and with whom. (2) Users can choose to share existing data that they have collected passively, to share prospectively, collect data, or both. For prospective studies, researchers can invite specific users to collect selected data during a specific time period. Users can also be notified about future studies by signing up for the research program. (3) Expiration dates are connected to each consent choice, which ensures that a user reconsiders his decision. A default expiration date of one year will be assigned, but users may also select personal expiration dates, such as an expiration date connected to the duration of the study. Users can choose to quit sharing data at any time, as required by GDPR regulations. During all steps, users are informed about implications of consent options. 

In the United States, the Sync for Science (S4S) [36] collaboration between Electronic Health Record (EHR) vendors, the National Institutes of Health (NIH), the Office of the National Coordinator for Health IT (ONC), and Harvard Medical School’s Department of Biomedical Informatics was started in 2016. S4S allows individuals to access their health data and share these data with researchers to support studies that generate insights into human health and disease [37]. Different EHRs collect and store health data differently, so S4S has focused on promoting both authorization and healthcare data standards to make it possible for EHR systems to release, upon patient approval, high quality data that researchers can readily consume. The All of Us Research Program [38] was the first study to adopt S4S technology in a pilot program. The program began national enrollment in 2018 and is expected to last at least 10 years. An initiative like Sync for Science gives the power to the patient: the patient can decide what information to share with researchers, and under what conditions. Much like many countries now have organ donation registration systems in place, this ‘data donation registration’ might be something that will be implemented around the world in the near future.

In the EU, the 1+ Million Genomes Initiative [39] is a good example of how many datasets could be combined into one large database, enabling the study of, e.g., rare diseases. The declaration aims to bring together fragmented infrastructure and expertise supporting a shared and tangible goal: one million genomes accessible in the EU by 2022. The 22 participating European countries envision that the digital transformation of genomic medicine (and healthcare in general) will help health systems to meet the challenges they face and become more sustainable, thereby improving the provision of high-quality health services and the effectiveness of treatments. They believe that this requires a concerted effort to overcome data silos, lack of interoperability and fragmentation of initiatives across the EU. Another recent example of such a large-scale collaborative data sharing effort is the Pan-Cancer Analysis of Whole Genomes (PCAWG) [40], which was facilitated by international data sharing using compute clouds. PCAWG contains 2658 whole-cancer genomes and their matching normal tissues across 38 tumour types. More than 1300 scientists and clinicians from 37 countries were involved in the project.

Data sharing goes beyond the academic world. Many public-private partnerships have been set up, in order to make sure that discoveries are not only published, but also applied in a product such as a medical device, a medicine or a computer program. Funding programmes, such as the Horizon 2020 Research and Innovation Programme of the European Union, very much stimulate data sharing with companies. In May 2016, it was announced that Deepmind, a company owned by Google and most famous for its innovative use of AI, was given access to the healthcare data of up to 1.6 million patients from three hospitals run by a major London NHS trust [41]. And in January 2018, Apple announced that they created a pact with 13 prominent health systems, including prestigious centers like Johns Hopkins and the University of Pennsylvania, allowing Apple to download the electronic health data of patients onto its devices, with consent [42]. Of course, when sharing data with commercial parties, privacy needs to be taken into account. For example, if GDPR applies, the patient needs to ‘opt-in’ for sharing their data for commercial use. Besides industry using data generated by academia, the opposite is also possible; these collaborations are called “data collaboratives” [43]. Data collaboratives are a new form of partnership in which privately held data are made accessible for analysis and use by external parties working in the public interest. By having researchers from both industry and academia work on the data, new insights and innovations can be created, and the potential of privately-owned data can be unlocked.

## 6. Federated Data

Data federation is a recent development in medical science, which is a possible solution for the data sharing vs. patient privacy dilemma. In this section, three examples of federated data systems for sharing medical data are discussed, resulting from the literature analysis from the introduction.

When applying machine learning methods on healthcare data, large samples sizes are required. Often these sizes can only be achieved by combining data from several studies. But this kind of pooling of information is difficult because of patient privacy and data protection needs. Privacy preserving distributed learning technology has the potential to overcome these limitations. The general idea behind distributed learning is that sites share a statistical model and its parameters, instead of sharing sensitive data. Each site runs computations on a local data store that generate these aggregated statistics. In this setting, organizations can collaborate by exchanging aggregated data/statistics while keeping the underlying data safely on site and undisclosed. VANTAGE6 [44] provides a way to use distributed learning technology, using open source software. It is one of the federated data systems that has recently become available in order to share data while preserving the patients’ privacy rights.

The Personal Health Train (PHT) [45,46] is another example of a federated data system; it aims to connect distributed health data and create value by increasing the use of existing health data for citizens, healthcare, and scientific research. The key concept in the Personal Health Train is to bring algorithms (‘trains’) to the data where they happen to be (‘stations’), rather than bringing all data together in a central database. The Personal Health Train is designed to give controlled access to heterogeneous data sources, while ensuring privacy protection and maximum engagement of individual subjects. As a prerequisite, health data are made FAIR (Findable, Accessible, Interoperable and Reusable) [33]. Stations containing FAIR data may be controlled by individuals, (general) physicians, biobanks, hospitals and public or private data repositories. The Personal Health Train was applied recently to a project with 20,000+ lung cancer patients [47] and will also be used in the Coronary ARtery disease: Risk estimations and Interventions for prevention and EaRly detection (CARRIER) project [48]. Likely more projects will follow.

Another example of a federated data system is DataSHIELD [49]. It provides a novel technological solution that can circumvent some of the most basic challenges in facilitating the access of researchers and other healthcare professionals to individual-level data. It facilitates research in settings where sharing the data itself is not possible (due to government restrictions, intellectual property issues, or data size). Commands are sent from a central analysis computer (AC) to several data computers (DCs) storing the data to be co-analysed. The data sets are analysed simultaneously but in parallel. The separate parallelized analyses are linked by non-disclosive summary statistics and commands transmitted back and forth between the DCs and the AC. DataSHIELD has been used by the Healthy Obese Project and the Environmental Core Project of the Biobank Standardisation and Harmonisation for Research Excellence in the European Union (BioSHaRE-EU [50]) for the federated analysis of 10 datasets across eight European countries.

## 7. Conclusions

This review discusses the current state of data sharing in healthcare. It shows that data sharing is widely supported by governments, funding programs and publishers, but that there are also issues. Clinicians or researchers might be reluctant to share data, because of publication pressure and fear for competition. This might be solved by the “open science” initiatives mentioned in this paper, which need to be supported by governments as well as the scientific communities itself in order to make it a success. Next to this, there are also patient-related issues such as stricter privacy laws. A possible (technical) solution here is the use of federated data systems such as the Personal Health Train, which enable algorithms to reach out the data without having the need to bring all data together. The challenges around privacy might also be solved by non-technical means such as using standardized consent forms to enable future use of data for research and/or commercial purposes. For the (near) future, there are some more developments that might be influential on the acceptance of open science and the sharing of data. One of these developments is medical crowdsourcing [51], which offers hope to patients who suffer from complex health conditions or rare diseases that are difficult to diagnose. Medical crowdsourcing platforms empower patients to use the “wisdom of the crowd” by providing access to a large pool of diverse medical information. One example medical crowdsourcing platform is CrowdMed [52]. This platform was appreciated by some patients with undiagnosed illnesses, because they received helpful guidance from crowdsourcing their diagnoses during their difficult diagnostic journeys [53]. Greater participation in crowdsourcing increases the likelihood of encountering a correct solution, and this might help to encourage patients to share their data. However, more participation can also lead to more noise, making the identification of the most likely solution from a broader pool of recommendations difficult. The challenge for medical crowdsourcing platforms is to increase participation of both patients and solution providers, while simultaneously increasing the efficacy and accuracy of solutions. Moreover, caution should be taken when giving people without a medical background the power to diagnose others. Another future development is the increase in “data generalists”; experts that focus entirely on data sharing and communication [54]. A data generalist takes on all responsibility for the sharing of data and needs critical thinking skills to integrate, evaluate and communicate the benefits and drawbacks of providing open data. They also have a role in data analysis. The emergence of this role should encourage better sharing of data. In a time where much attention is going to the data scientist, it could be the data generalist that really has the job of the future.

## Figures and Tables

**Figure 1 ijerph-17-03046-f001:**
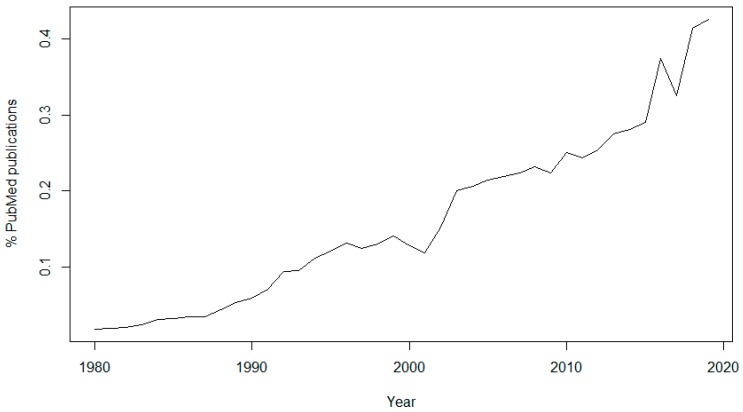
Graph of the number of abstracts of PubMed publications containing the keyword “data sharing” as a percentage of the total, per year since 1980.

**Figure 2 ijerph-17-03046-f002:**
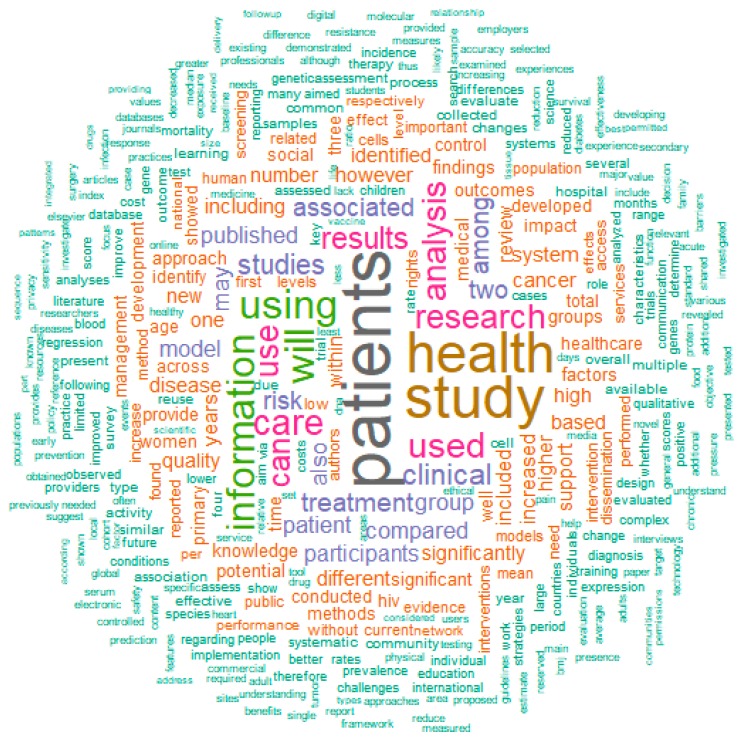
Wordcloud of all abstracts of PubMed publications containing the keyword “data sharing”, generated by the R package PubMedWordcloud [13].

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
