# Peer review of "Sharing Is Caring—Data Sharing Initiatives in Healthcare"

_ijerph, 2020, doi:10.3390/ijerph17093046_

Round 1

Reviewer 1 Report

please see the comments in attached file.

Reviewer 2 Report

The author discusses a growing issue in the realm of research. He identifies several initiatives under way as several proposed solutions.  One of the problems is that the author has a tendency to use "they must reference..", "they require investigators...".  It is not clear who the "They" are and this needs to be clarified each time the phrase "They....." is used.  In reading the paper, it is not clear what is "proposed" and what is actually happening.  Several places the author suggests "proposals" but then treats them as though they are "in existence" already. For example, in the discussion of the ICMJE, the author looks at the "proposed" solution.  Is this in effect?  Or just a proposal?  The author needs to point out that "reproducibility" is partially a result of merging several data sets that have operationalized the key variables in different ways.  Thus, "coding" issues are problematic when trying to use "merged" data--beyond what the author talks about.  Finally, I was surprised that the author didn't talk more about the American and Canadian ethics guidelines relevant to data sharing.  And, I should point out that in the last paragraph of section 5 (second sentence), the sentence does not make any sense.
